# Emergent Reciprocity and Team Formation from Randomized Uncertain Social Preferences

**Bowen Baker**
OpenAI
bowen@openai.com

## Abstract

Multi-agent reinforcement learning (MARL) has shown recent success in increasingly complex fixed-team zero-sum environments. However, the real world is not zero-sum nor does it have fixed teams; humans face numerous social dilemmas and must learn when to cooperate and when to compete. To successfully deploy agents into the human world, it may be important that they be able to understand and help in our conflicts. Unfortunately, selfish MARL agents typically fail when faced with social dilemmas. In this work, we show evidence of emergent direct reciprocity, indirect reciprocity and reputation, and team formation when training agents with randomized uncertain social preferences (RUSP), a novel environment augmentation that expands the distribution of environments agents play in. RUSP is generic and scalable; it can be applied to any multi-agent environment without changing the original underlying game dynamics or objectives. In particular, we show that with RUSP these behaviors can emerge and lead to higher social welfare equilibria in both classic abstract social dilemmas like Iterated Prisoner's Dilemma as well in more complex intertemporal environments.

## 1 Introduction

Many real world problems require complex coordination between multiple agents, and multi-agent reinforcement learning (MARL) has recently shown great success in impressive two-team zero-sum settings such as Go,[1] DoTA,[2] Starcraft,[3] hide-and-seek,[4] and capture the flag.[5] Zero-sum two-player (or two-team) settings are convenient yardsticks for progress in complex control, for there exists a Nash equilibrium which coincides with the minimax solution.[6] However, the human world is far messier. We constantly face social dilemmas at multiple scales, from the interpersonal to the international, and we must decide not only how to cooperate but *when* to cooperate.

Can we not simply train purely prosocial agents that are always cooperative with humans? We argue we cannot, for the agents we deploy must be robust to a world in which there will be uncooperative humans and uncooperative agents trained by either malicious or careless actors. If agents only ever see cooperation during training, any defection will be out of distribution, and therefore we will likely have no guarantees on their behavior. Transfer to the human world aside, the social intelligence hypothesis posits that the additional cognitive tasks organisms must perform due to social dilemmas, norms, and structures created additional pressure for intelligence past that of base survival,[7,8] and so training in mixed cooperative-competitive settings are of further independent interest in that they could be key to creating increasingly intelligent agents.

Unfortunately, the MARL methods that have been so successful in zero-sum two-team settings have historically failed in social dilemma environments[9,10,11,12] despite the existence of more cooperative equilibria with higher social welfare. There has been much work in augmenting learning agents to arrive at more cooperative equilibria in social dilemma matrix games[9,13,14,15] and more complex environments;[12,16,17,18,19,20,21,22,23] however, few among these have led to robust cooperation such

| Player 2 \ Player 1 | Cooperate | Defect |
|---|---|---|
| **Cooperate** | R / R | S / T |
| **Defect** | T / S | P / P |

Social Dilemma Conditions

| | | |
|---|---|---|
| 1. | R > P | Mutual cooperation (CC) is preferred to mutual defection (DD) |
| 2. | R > S | Mutual cooperation (CC) is preferred to unilateral cooperation (CD) |
| 3. | 2R > T + S | Mutual cooperation (CC) is preferred to rotating unilateral cooperation and defection (CD/DC) |
| 4. | T > R | Greed: Unilateral defection (DC) is preferred to mutual cooperation (CC) |
| | and/or P > S | Fear: Mutual defection (DD) is preferred to unilateral cooperation (CD) |

Figure 1: **Matrix Game Social Dilemmas.** In their simplest form, players have two actions: Co-operate (C) and Defect (D), and receive payouts dependent on their joint actions. Following the formulation in Macy and Flache[25] and Leibo et al.[12], players receive $R$ (reward) and $P$ (punishment) for mutual cooperation and defection, respectively, whereas players receive $S$ (sucker) and $T$ (temptation) for one agent cooperating and one defecting. The three canonical games, Chicken, Stag Hunt, and Prisoner's Dilemma, are differentiated by the incentive to defect. In Chicken, we have $T > R$ meaning players are tempted to defect out of *greed*. In Stag Hunt, $P > S$ which pressures players to defect out of *fear*. In Prisoner's Dilemmas, the most difficult scenario, players are motivated by both greed and fear.

as reciprocity, and many require changing underlying environment dynamics, agent capabilities, or agent objectives.

In this work we propose to train agents with randomized uncertain social preferences (RUSP), a novel multi-agent environment augmentation that *expands* the distribution of environments in which agents train. During training, agents share varying amounts of reward with each other; however, each agent has an independent degree of uncertainty over their relationship with another, creating information asymmetry that we hypothesize pressures agents to learn socially reactive policies. Prior works have also used prosociality during MARL training,[24,20] though RUSP differs in a number of ways:

1. We fully randomize the reward transformation matrix rather than using a fixed linear combination of selfish and social welfare.
2. Agent relationships are uncertain, creating explicit information asymmetry.
3. We condition agents on their relationships rather than holding them fixed per population, allowing us to evaluate agents in the original game without any social preferences.

The main contributions of this work are: 1) We propose a novel multi-agent environment augmentation, RUSP, that pressures agents into more cooperative equilibria and extensively evaluate it in social dilemmas embedded in matrix games, grid-worlds, and physics-based worlds. 2) This is the first method to our knowledge that has given rise to both emergent reciprocity and team formation in MARL agents. 3) We open-source our environments for further research into social dilemmas.[1]

## 2   Preliminaries

**Social Dilemmas.** Mixed motive iterated matrix games have provided grounds for research into social dilemmas for decades.[26] In the finite horizon Iterated Prisoner's Dilemma (IPD) – see Figure 1 – where players play many consecutive rounds, the only Nash equilibrium is all-defect. However, the Folk theorem tells us that in infinite horizon IPD there are infinitely many equilibria including popularly known cooperative strategies such as Grim Trigger and Tit-for-Tat for particular discount rates.

Despite the existence of cooperative equilibria with both higher individual and social welfare, it is commonly known that reinforcement learning methods often fail to find these.[10,11] It has been shown that reciprocal strategies are evolutionarily stable,[27,28] but works like these often leave open the question of how to generate these strategies in complex domains. The notion of sequential social dilemmas[12] was recently proposed to begin studying cooperation in environments more reminiscent of the real world; instead of being atomic actions, cooperation and defection can be potentially long sequences. In these domains, there has been success in finding cooperative and reciprocal strategies using second order optimization;[22,29] however, these methods are likely to be

prohibitively expensive for large models and numbers of agents. Others have incentivized cooperation through inequity aversion,[17] intrinsic motivation for social influence,[19] and rewarding agents for reciprocal behavior.[20] Still others have learned individual cooperative and competitive sub-policies, choosing between them with a high level reciprocal controller.[30,16] Most similar to our work, it has been shown that including prosocial agents in the population can lead to higher social welfare outcomes,[24,20] though they do not show the emergence of reciprocity.

**Team Formation.** In this work, we consider the problem of team formation a social dilemma: how first should potentially symmetric learning agents break symmetry and form sustained coalitions in the face of temptation to "backstab" a teammate for short term gains? There has been much prior work in cooperative game theory on coalition formation[31] such as how to divide rewards within a coalition,[32,33,34] how to find coalitions in cooperative graphs,[35] and how to negotiate or form contracts.[36] Recent work in MARL finds that agents can learn cooperative behavior when augmenting agents with explicit contract mechanisms over future actions;[37,21] however, defining contracts over actions in complex environments where cooperation requires a long sequence of operations may be difficult, and in this work we aim to show emergence of team formation without this requirement.

**Payoff Uncertainty.** We are not the first to study uncertainty over agent payoffs. Of particular relevance, Kreps et al.[38] show that in the finite horizon iterated prisoner's dilemma, where normally pure defection is the only equilibrium, agents will cooperate often under the presence of either uncertainty over the other agent's strategy, i.e. it is apriori unknown whether the opponent is rational or playing tit-for-tat, or uncertainty over the opponent's payoffs, i.e. it is apriori unknown whether the opponent has a preference for cooperation. These sequential equilibrium models[39] were later experimentally validated in human trials.[40] There has also been work showing that uncertainty over payoffs is critical to the value alignment problem inherent in agent-human collaboration.[41]

# 3 Motivation and Method

A reinforcement learning (RL) agent trained against an adaptive strategy such as tit-for-tat will learn to be cooperative until the last timestep in fixed horizon social dilemmas and indefinitely in infinite horizon games.[9] However, learning agents pitted against each other, rather than against fixed strategies, often converge to mutual defection[9,10,11] despite the existence of higher social welfare Nash equilibria.[42,27] Downstream reciprocal strategies such as tit-for-tat are *reactive* strategies; they follow or adapt to the behavior of their opponent. Intuitively, uncertainty over opponent strategy may also lead to reactive behavior; for instance, if you train a RL agent against 50% all-defect and 50% tit-for-tat strategies in IPD, the RL agent will follow the lead of its opponent, defecting and cooperating respectively. However, if myopic RL agents often converge to unimodal defective strategies, then they will never see this necessary variance during self-play training.

One way to induce uncertainty into opponent strategy is introduce game dynamics that create *information asymmetry* between agents. For instance, take a cooperative game where agents are rewarded for completing a communal task $\mathcal{T}$ which is drawn at random at the beginning of the episode. If all agents know $\mathcal{T}$ and also who their teammates are, then they can execute a strategy without any knowledge of teammate actions (assuming a deterministic environment). However, if only one agent is given the communal task, other agents must follow the lead of that agent, i.e. be reactive, as they have no information over $\mathcal{T}$.

Many games do not naturally have tasks or other variables over which we can induce asymmetric uncertainty. However, because randomizing tasks is equivalent to randomizing reward functions, a suitable set of generic reward transformations would admit the ability to induce asymmetric uncertainty in even a fixed payout game like the classic Prisoner's Dilemma. Reward sharing transformations are commonly used to solve social dilemmas[24,17,20] and can be generically applied to any multi-agent game. We hypothesize that if we induce asymmetric uncertainty over agents' social relationships they will learn *socially reactive* behavior. Just as if you do not know the task at hand and must react to your partners actions, if you do not know if another agent is friend or foe, you must react to how cooperative they are. To this end, we have agents condition on a noisy observation of the reward transformation with independent levels of uncertainty. While prior work has kept reward transformations fixed per population, we sample a new reward sharing transformation

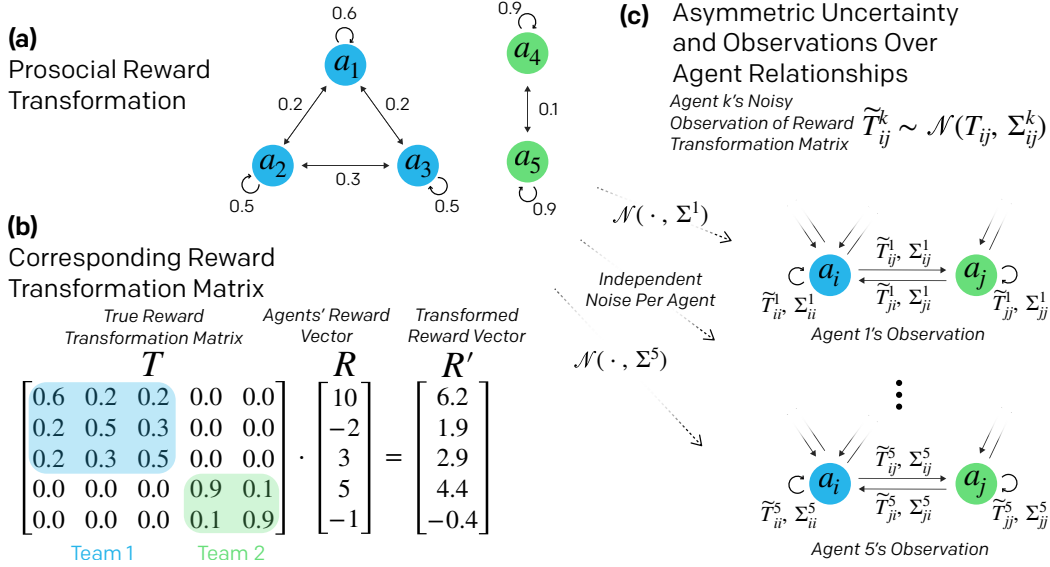

Figure 2: **Randomized Uncertain Social Preferences (RUSP).** (a) Agents are first partitioned into groups, which are independent cliques in the agent relationship graph. Here we show two groups; however, in general we sample uniformly from the set of integer partitions. Edge magnitudes are sampled independently and represent the relative amount the connected agents share reward with each other. (b) This graph structure is equivalent to having block-sparse row normalized reward transformation matrix. (c) Agents do not directly observe their social relationships but rather a noisy variant. For each agent-agent pair $(i, j)$, agent $k$ receives a noisy observation over their relationship $\widetilde{T}_{ij}^{k}$ along with its uncertainty over that observation $\Sigma_{ij}^{k}$, which is also sampled independently per agent $k$ and per relationship $(i, j)$.

per episode and condition agents on their relationships such that we can still evaluate them in the fully selfish, original game.

In particular, during each episode agents are partitioned on to randomized *soft* teams, meaning that they share rewards but may prioritize teammates more or less than themselves, rather than *hard* teams with completely shared rewards. Rather than completely randomizing the matrix, we hypothesize including episodes with sparse relationship graphs will cause agents to learn to partition better into coalitions, which we will show preliminary evidence is true. As seen in Figure 2(b), this manifests as a block sparse row normalized reward transformation matrix, $T$. Agent $k$ observes elements of the reward transformation matrix with added noise, $\widetilde{T}_{ij}^{k} \sim \mathcal{N}(T_{ij}, \Sigma_{ij}^{k})$. Agents' uncertainty over their relationships are sampled such that agents have asymmetric information, i.e. $\Sigma_{ij}^{k}$ is sampled independently per agent $k$ and per relationship $(i, j)$. Agents observe their own level of uncertainty along with their noisy samples from $T$.

Importantly, training with randomized uncertain social preferences only *expands* the distribution of environments agents are trained in. The original unmodified selfish environment — no social preferences ($T = I$) and no uncertainty ($\Sigma = 0$) — is within the RUSP distribution, meaning we can evaluate agents in the original selfish setting. To our knowledge this is a novel approach to training agents with randomized social preferences, where prior works hold social preferences static.[24,17,23,20]

In this work we partition agents by uniformly sampling from all unique integer partitions, including singleton teams. For example, in Figure 2(b) agents have been partitioned into cliques of size 3 and 2; however, all combinations are possible. Within each clique, relative reward sharing values (non-zero entries in the reward transformation matrix $T$) are sampled from a uniform distribution $U[0, 1]$ after which $T$ is row normalized. This work only considers prosocial transformations; however, in general antisocial transformations (allowing elements of $T$ to be negative) are also possible. Uncertainty levels are sampled independently per agent $\Sigma_{ij}^{k} \sim U[0, \sigma_{max}]$. These choices were rather arbitrary;

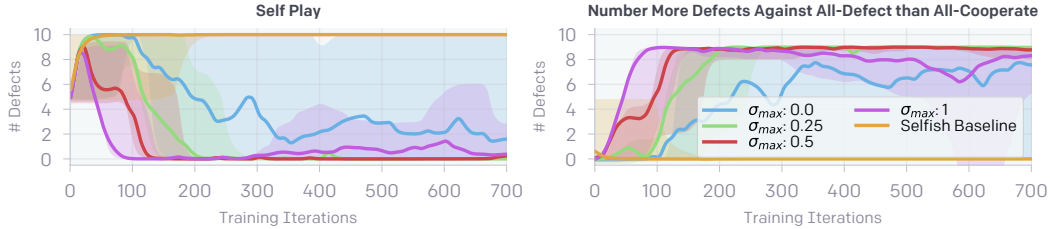

Figure 3: **Reciprocity in Infinite Prisoner's Dilemma.** Agents' uncertainty are independently sampled from $U[0, \sigma_{max}]$ during training; however, it is important to remember that when evaluating agents they are given no prosocial preferences or uncertainty ($T = I$, $\Sigma = 0$). During training, the episode ends with a probability 0.1 (mean horizon of 10) at each timestep; during evaluation we fix the horizon to be 10 timesteps exactly. (Left) We show the number of defect actions the agent makes against itself versus training iteration, showing the resultant equilibrium. (Right) We play the trained agent against all-defect and all-cooperate policies and compare the number of defect actions taken against each. A higher value here is evidence of a more reciprocal strategy. For all experiments including this, we report the mean over 10 independent runs and shade the 90% confidence interval.

they required minimal tuning, and we hold the hyperparameters constant along all experiments and domains.

For all experiments, agent policies are recurrent entity-invariant neural networks similar to Baker et al.[4] trained with proximal policy optimization (PPO),[43] an on-policy reinforcement learning algorithm; see Appendix C for more details on the policy architecture and policy optimization. For all plots, one training iteration comprises 60 steps of stochastic gradient descent on the PPO objective.

# 4   Emergent Reciprocity

Both direct and indirect reciprocity have been identified as mechanisms that lead to sustained cooperation in humans.[44] We start by investigating the classic infinite horizon Iterated Prisoner's Dilemma (IPD). We set $R = 2$, $P = 0$, $S = -2$, and $T = 4$; see Figure 1 for details on the IPD criteria. Episode lengths are sampled from a geometric distribution with stopping probability 0.1, which is equivalent to an infinite horizon game with discount factor $\gamma = 0.9$ with a mean horizon of 10. Agents observe the last action both they and their opponent took as well as the current time-step, but do not observe the horizon for that episode.

## 4.1   Direct Reciprocity

We begin with a simple 2-agent case (classic IPD). During training each agent has their uncertainty level independently sampled from $U[0, \sigma_{max}]$, and we compare this to a selfish baseline with no randomized social preferences or uncertainty. At evaluation, all agents are given no social preferences or uncertainty, i.e. $T = I$ and $\Sigma = 0$, meaning they are playing the original IPD game with no modifications.

Figure 3 shows the effect of training in IPD with randomized social preferences and varying levels of uncertainty. First, we show the number of defect actions when the trained agents play against themselves (self-play), giving insight into the resultant equilibrium. As expected, the selfish baseline converges to the all-defect equilibrium; however, we find that agents trained with RUSP find cooperative equilibria, and in particular we see that a higher uncertainty limit (higher $\sigma_{max}$) during training leads agents to find cooperative equilibria more quickly. However, creating cooperative agents is not enough for they could be naively cooperating and exploitable in the face of defectors. To differentiate whether our agents have learned either reciprocal or naively cooperative strategies we compare the number of defects they make against fixed all-defect and all-cooperate policies. We find that RUSP agents are indeed reciprocal and will punish players for defecting against them. Here we see that training with non-zero uncertainty is important to the stability of emergent reciprocity; in the Appendix A we also ablated the effect of information asymmetry and playing against past policy versions[45] during training and found both were crucial.

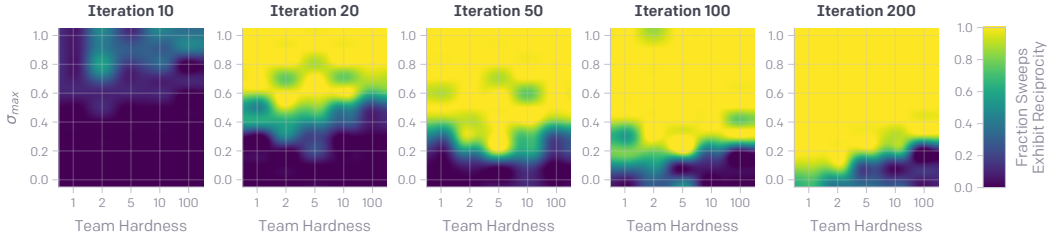

Figure 4: **Reciprocity and Social Preference "Hardness" in IPD.** Instead of sampling social preferences $T_{ij}$ from a uniform distribution, we sample it from a $\beta$-distribution. We hold $\beta = 1$ fixed and vary $\alpha$ from 1 to 100, where $\alpha = 1$ is the same as the uniform case or "soft" social preferences explored in this work and $\alpha = 100$ is the "hard" social preferences case. Here we label a policy reciprocal if it defects against an all-defect policy at least 1 more time than against an all-cooperate policy. We find that softer social preferences and higher $\sigma_{max}$ leads to reciprocal equilibria faster, and notably when training with hard social preferences and no noise, reciprocity never emerges.

*Hard* teams, where agents share reward equally, are most often investigated in multi-agent environments.[2,3,4,5] In Figure 4 we show how emergence of reciprocity progresses through training as a function of the hardness of agents' social preferences and their maximum uncertainty value, $\sigma_{max}$. We find that higher uncertainty leads to faster emergence of reciprocity, which is consistent with the results shown in Figure 3, and most notably reciprocity never emerges when training with hard preferences and no uncertainty.

## 4.2 Indirect Reciprocity and Emergent Reputations

"Indirect reciprocity involves reputation and status, and results in everyone in the group continually being assessed and re-assessed." – Richard Alexander.[46] Not only has indirect reciprocity been hypothesized to be a mechanism leading to sustained cooperation in humans,[44] but it has also been suggested it could be a pressure for increasing cognitive complexity.[47]

In this work, we have a group of agents (in this case 3) randomly take turns playing Prisoner's Dilemma where all agents can observe the actions taken by those playing. We use the same payout matrix as in Section 4.1 but double the mean horizon to 20. To measure whether our agents have learned indirectly reciprocal strategies and reputation tracking, we evaluate them in two additional settings described below and in Figure 5.

**Evaluation Setting: *Hold Out*.** To identify whether agents have indeed learned to keep track of other agents' behavior, we replace one of the three agents with either a fixed all-defect or all-cooperate policy, and we hold the agent being evaluated out of the game until the last timestep at which point it plays against the fixed policy for the first time. We show results in Figure 5 where we measure how often the trained agent defects against the all-defect policy as compared to the all-cooperate policy. We find that RUSP agents do indeed learn to condition on prior games of other agents, as they much more often than not reciprocate opponent behavior even though they have never played against each other before.

**Evaluation Setting: *Prior Rapport*.** The *Hold Out* game evaluates whether agents can condition on prior history, but it does not explicitly tell us if agents have learned to track the reputation of others over many timesteps, discriminately cooperating and defecting. We further investigated by first playing the agent under question $a_1$ against another RUSP agent $a_2$ for a few timesteps such that they could potentially gain rapport, then playing $a_2$ against an all-defect policy $a_D$, and finally playing $a_1$ against both other agents. Perfect play here should be to cooperate with $a_2$ and defect against $a_D$, and we found that indeed $a_1$ would cooperate with $a_2$ more than $a_D$ but by no means perfectly (up to 60% more in some seeds). We found this drastically improved with higher amounts of past policy play during training (see Appendix A for an ablation of past-play in 2-player IPD), indicating that potentially inducing more variance in opponent play could improve emergence of reputation further; however, we leave this to future work.

Prior work has shown that policies endowed with indirect reciprocal strategies and reputation mechanisms are evolutionarily stable under certain conditions.[48,49] Our agents have no such built in

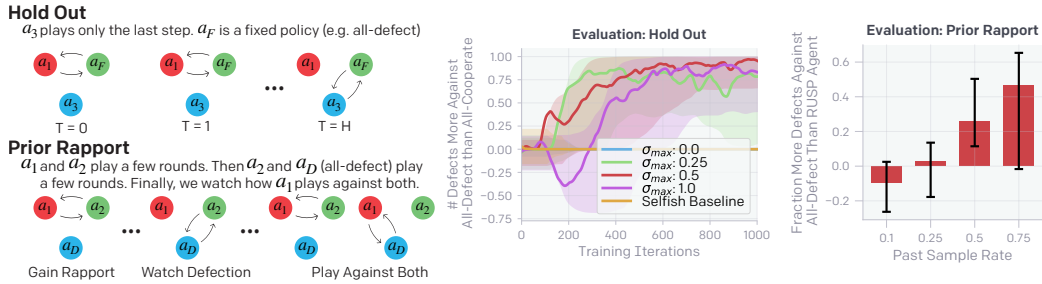

Figure 5: **Indirect Reciprocity in Iterated Prisoners Dilemma.** During training, two of the three agents are randomly sampled on each timestep to play Prisoner's Dilemma. (left) We show the two settings in which we evaluate agents. (middle) We compare how often RUSP agents defect against the fixed agent in the *Hold Out* setting. Notably, both the selfish baseline and RUSP agent trained with no uncertainty failed to ever learn any form of reciprocity in this setting. (right) In the *Prior Rapport* setting, we measure how often a RUSP policy cooperates with another cooperative policy after they've been able to gain rapport. We find that only RUSP agents trained with high amounts of past policy sampling and non-zero uncertainty learn any form of reputation system (here we show RUSP agents with $\sigma_{max} = 0.5$ after 1000 training iterations). A past sample rate of 0.1 indicates a 10% chance of an agent playing a policy uniformly sampled from all past versions only during training. At evaluation both $a_1$ and $a_2$ use the latest policy version.

mechanisms but rather are parameterized with recurrent neural networks, indicating that they learn to keep track of reputation in learned memory activations. To our knowledge this is the first work showing evidence of an *emergent* reputation system and indirect reciprocity in social dilemma games from learning based agents.

# 5 Emergent Team Formation

Team formation is a hallmark behavior of both humans and animals.[50] We form coalitions at multiple scales, from study groups to nations, and rather than being fixed they change over time; defectors are exiled and new entities are given a chance to join. We first show team formation results in an abstract environment we call *Prisoner's Buddy*, where agents must form coalitions and resist the temptation to back-stab their teammate, and then we show initial results in a complex physics based world, *Oasis*.

## 5.1 Prisoner's Buddy

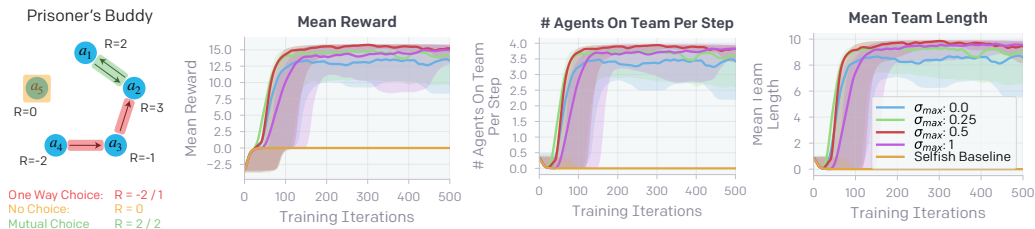

Figure 6: **Team Formation in Prisoner's Buddy.** (Left) We show example agent actions depicted with arrows and rewards written next to each agent, e.g. agent $a_4$ has chosen agent $a_3$ and has received a reward of -2. An agent is rewarded if others choose it and punished if its own choice is not reciprocated. (Right) We show the mean reward across agents, the number of agents that have found a buddy (are on a team) at each step, and finally the mean team length of any coalition that formed during the episode (mean number of consecutive steps two agents choose each other).

In *Prisoner's Buddy* agents receive reward by "finding a buddy". On each timestep, agents act by either choosing another agent (via an attention mechanism) or deciding to choose no one and sitting out that round. If two agents mutually choose each other, they each get a reward of 2. If an agent Alice chooses Bob but the choice isn't reciprocated, Alice receives -2 and Bob receives 1. Finally, if

an agent chooses no one, they receive 0. Figure 6 (left) depicts an example round of Prisoner's Buddy. The two-player version of this game is a stag hunt; however, with more than two players there can exist a Prisoner's Dilemma. For instance, if Alice and Bob have formed an alliance, another agent Eve can tempt Alice away from that alliance. If Bob doesn't change strategy, then Alice can achieve more reward by forming a new alliance with Eve while Bob makes an unreciprocated choice of Alice.

We see in Figure 6 (right) that selfish MARL agents fail to form any teams and achieve 0 reward in 5-player Prisoner's Buddy. We also see that not only does RUSP cause agents to successfully partition into teams, but we find that those teams that are initially formed are often stable and maintained throughout the episode, which is indicated by the mean team length nearing the episode horizon. This ability to successfully break symmetry and maintain sustained teams of two is more stable with higher uncertainty, and in Appendix A we additionally verify that integer partitioning as opposed to fully randomizing the transformation matrix during training also leads to more stable team formation.

## 5.2 Oasis

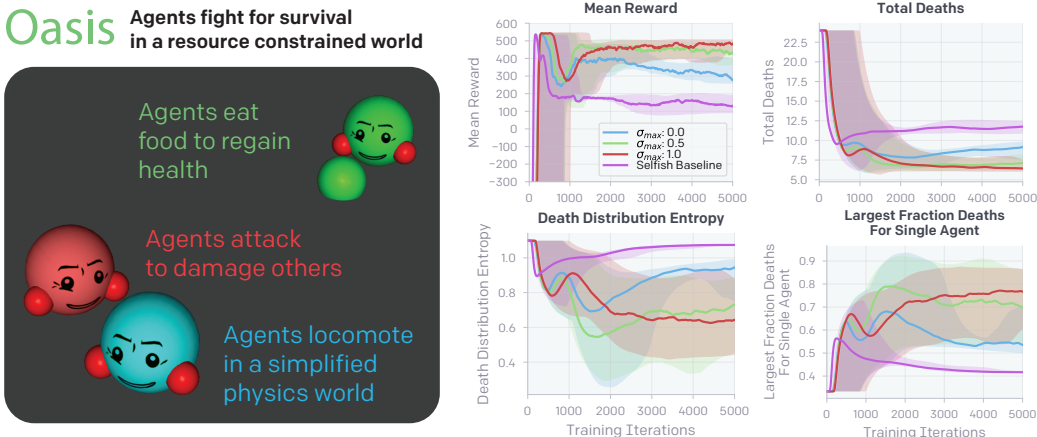

Figure 7: **Team Formation in Oasis.** (Left) We show a rendering of the *oasis* environment. (Right) We show mean reward of agents, the total number of deaths per episode, the entropy of the death distribution (see text for details), and the largest fraction of total deaths attributed to any individual agent.

We next show preliminary team formation results in a much more complex environment. Here cooperation is no longer an atomic operation, but a potentially long and complex sequence of actions much more reminiscent of the human world. *Oasis* is a MUJOCO[51] physics-based environment where agents are tasked with survival; their only reward is +1 for every timestep they remain alive and a large negative reward when they die. Their health decreases with each step, but they can regain health by eating food pellets and can attack others to reduce target health. If an agent is reduced below 0 health, it dies and respawns at the edge of the play area after 100 timesteps. The environment is resource constrained; there is only enough food to support two of the three agents.

The constraint over resources creates a social dilemma. First, two agents need to break symmetry and gang up on the third to secure the food source and stay alive. Secondly, let's assume that agents Alice and Bob were able to successfully break symmetry and gang up on agent Eve, but in the process Eve attacked Bob injuring him. Should Alice turn on Bob in favor of Eve after Eve respawns with full health or should Alice maintain the coalition with Bob even though he is currently weak and gang up on Eve again? This is very similar to the social dilemma in Prisoner's Buddy where there is short term incentive to turn on a teammate you had in the pass, but there is long term benefit to always being on a team.

We show results when training with and without RUSP in Figure 7 in a 3-player game. Specifically, we show the mean reward, the total number of deaths, the entropy of the death distribution (i.e. $H = -\sum p_D^i \log(p_D^i)$ where $p_D^i$ is the empirical probability within an episode that a death will be attributed to agent $i$), and the largest fraction of deaths attributed to any single agent during an episode. We first see that the game is indeed a social dilemma. The mean reward is lower and the

number of deaths higher for the selfish baselines as compared to RUSP agents. We also see that the entropy of the death distribution is highest for the selfish baseline (deaths are fairly evenly distributed across all agents during an episode), which is confirmed by the largest fraction of deaths attributed to a single agent being closest to 1/3. These measures indicate that the selfish baseline agents have failed to break symmetry and cooperate.

We also see that RUSP agents perform much better than the selfish baseline: they achieve higher reward and die less frequently. Non-zero uncertainty over relationships also causes agents to both find more efficient equilibria and exhibit signs of stronger team formation than both the selfish and fully certain ($\sigma_{\max} = 0$) baselines. We see that as agents train, the entropy of the death distribution diminishes and the fraction of deaths allocated to a single agent increases. In fact, for agents trained with uncertainty levels up to $\sigma_{\max} = 1.0$ we find that up to 90% of the deaths in an episode will be attributed to a single agent, indicating that two agents have learned to form a coalition and mostly exclude the third from the food source.

While these results show promising signs of emergent team formation, there is clearly room for improvement. We independently verified that if trained with fixed teams of size 2 and 1, agents on the team can consistently control the food source and achieve higher reward than RUSP agents currently do, indicating that their inability to form perfect teams is due to a learning issue rather than inherent to the environment. Furthermore, agents take quite a while to reach policies that exhibit team formation; with our training setup in Oasis, 1000 iterations corresponds to roughly 3.8 million episodes of experience. This indicates the need to further investigate RUSP and make it more efficient in long horizon games where cooperation is no longer an atomic action as it is in simple matrix games.

# 6 Final Remarks

Reciprocity and team formation are hallmark behaviors of sustained cooperation in both animals and humans.[50,44] The foundations of many of our social structures are rooted in these basic behaviors and are even explicitly written into them — almost 4000 years ago reciprocal punishment was at the core of Hammurabi's code of laws. If we are to see the emergence of more complex social structures and norms, it seems a prudent first step to understanding how simple forms of reciprocity may develop in artificial agents.

In this work, we've demonstrated that training reinforcement learning agents with randomized uncertain social preferences can lead to emergence of both reciprocity and team formation even when the agents are evaluated with no social preferences. Our method is simple and scalable; it can be applied to any multi-agent environment, and we found that it required no modifications when moving from experiments in abstract matrix games to more complex environments. To further this point, we show results in the popular grid-world Cleanup (public goods) and Harvest (tragedy of the commons) environments proposed in Hughes et al.[17] in the Appendix and find that RUSP without modification is able to find cooperative equilibria in both games.

While we've seen moderate success with RUSP, our experiments with indirect reciprocity and Oasis show that RUSP as currently presented is not a silver bullet and there is much room to improve. Because RUSP can generally be applied to any multi-agent environment, it could be combined with other methods aimed at improving agent cooperation. Furthermore, we only experimented in environments with small numbers of agents; however, the credit assignment problem grows exponentially with number of agents in cooperative settings. We leave exploring RUSP's generalization to large numbers of agents to future work; however, reasonable avenues may include curricula slowly increasing the degree of reward sharing as in Berner et al.[2] or number of agents as in Long et al.[52] Finally, since the first version of this paper, we've experimented with antisocial preferences and found that they produced emergence of reputation much more stably, which in hindsight is unsurprising. We furthermore found that across most environments if agents are evaluated in the case where they are told they are uncertain $\Sigma = \sigma_{\max}$ but without any social preferences or noise ($T = \tilde{T} = I$) they generally perform better. We leave these anecdotes here to help direct potential future work into RUSP.

# 7 Acknowledgements and Disclosure of Funding

RUSP was initially born from conversations with David Farhi who also provided feedback on the final manuscript, so we thank him first and foremost for his insight and time. We also thank Bob McGrew, Ilya Sutskever, Glenn Powell, Matthias Plappert, Todor Markov, Yi Wu, Ingmar Kanitscheider, Jeff Clune, Adrien Ecoffet, Joost Huizinga, and various others within OpenAI for their comments on initial versions of RUSP and on the final manuscript.

Bowen Baker is a full-time employee of OpenAI from which all funding was provided.

# 8 Broader Impact

The human world is social; we often find ourselves in conflict at multiple scales, from our daily lives with a friend or colleague to the international stage. As we continue to deploy artificial agents into our world and give them increasing amounts of responsibility, it will be important that they understand our social dilemmas. Were we to train completely cooperative agents against each other, the notion of defection would never emerge, uncooperative behavior would be out of distribution, and we would have no guarantees on their behavior. For instance, if two entities with possibly misaligned objectives send artificial agents to negotiate a deal or collaborate on a project, those agents should be able to cooperate without being exploited. Similarly, say an entity sends a fully cooperative agent to collaborate with two humans who have misaligned objectives; that agent should expect potential uncooperative behavior from one of them and plan accordingly rather than assume all parties involved will be cooperative. In order for agents to generalize to a world with heterogeneous motives, they must see instances of mixed-motive interactions during training.

One very reasonable avenue for agents to gain this knowledge would be to collect enough real data from a variety of human social dilemmas, train agents on the solutions, and hope that they generalize to new social dilemmas in the future. The data collection path is likely to yield fruit in the short term, but learning from human collected data may have a limit. For instance, in the recent work producing super-human agents in Starcraft 2,[3] they directly compare agents trained solely with supervised data from human games to a combination of supervised learning and self-play, and they found the latter to be far superior.

Just as self-play and self-supervised learning processes have proven critical in training superhuman agents in challenging video games, they may also provide an avenue to producing agents superior to humans at solving our own social issues if paired with the right environments. However, as we've seen in this work and others, the naive multi-agent algorithms that have been extremely successful in zero-sum two-team settings dismally fail when confronted with social dilemmas, converging to all-defect equilibria.

Allowing agents to learn and choose when to defect and when to cooperate may bring a host of additional safety problems. For instance, a commonly known issue with the tit-for-tat strategy is that if agents make an error, they won't be able to recover and will defect forever. Making guarantees that agents are safe and making correct choices will be even more of an issue than with purely cooperative agents.

In this work we propose a generic method that leads to both reciprocity and team formation, hallmark behaviors of sustained human cooperation. While not directly relevant to current applications of artificial intelligence, we hope that this work in tandem with prior methods will lay the ground for future artificial agents to (1) have experience in solving social dilemmas similar to those in the human world and (2) endlessly learn and complexify from the pressure of social autocurricula.[53]

## Footnotes

[1]Environment code will be available at `github.com/openai/multi-agent-emergence-environments`

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
