[Supplementary Material]

# A    RUSP Ablations

Figure 8: **Information Asymmetry in Iterated Prisoner's Dilemma.** We present evidence in favor of our hypothesis that information asymmetry is an important factor in emergent reciprocity. Throughout the main paper, we saw that often higher levels of the maximum uncertainty were better; however, was it simply important to add noise to the agents' observations or does it have to be asymmetric noise? In blue we show the baseline RUSP method (asymmetric noise with uncertainties sampled independently from $U[0, \sigma_{max}]$). In green we show the case where now uncertainties and noise over relationships are not sampled independently, i.e. $\Sigma_{kj}^i = \Sigma_{kj}^\ell$ and $\widetilde{T}_{kj}^i = \widetilde{T}_{kj}^\ell$. We see that in this case, agents never learn reciprocal behavior and converge to the all-defect equilibrium. Next, in red we show the case where agents are given symmetric uncertainty, but asymmetric noisy samples from the transformation matrix, i.e. $\Sigma_{kj}^i = \Sigma_{kj}^\ell$ and $\widetilde{T}_{kj}^i \neq \widetilde{T}_{kj}^\ell$. In this case, we find that agents do indeed converge to cooperating with each other, but do not learn a strategy as strongly reciprocal as when trained with asymmetric uncertainty, indicating that they often naively cooperate even with an all-defect policy that has consistently taken them for a sucker.

Figure 9: **Past Play in Iterated Prisoner's Dilemma.** During training, in 10% of agents in the game do not use the latest policy weights but sample uniformly from past versions. First proposed as a double oracle algorithm,[45] it has become standard practice in multi-agent reinforcement learning.[2,3,4] We find that past play greatly stabilizes the emergence of reciprocity in IPD. We also found that it was crucial in the emergence of reputation in our indirect reciprocity experiments in Section 4.2.

Figure 10: **Integer Partitioning versus Full Matrix Randomization in Prisoner's Buddy.** Here we ablate the choice we made to sample from all integer partitions when constructing the reward transformation matrix rather than having the entire matrix randomized and non-zero. We hypothesized that in games like prisoner's buddy where teams of 2 are optimal, more often having clique structure during training could help agents learn to choose that structure when none is give. We cannot directly verify this is true; however, we do find that partitioning during training does indeed help the emergence of stable team formation.

## B  Public Goods and Tragedy of the Commons Grid-Worlds

The Harvest and Cleanup environments[17] (shown in Figure 11) are social dilemma environments embedded in a grid world. Here, cooperation and defection are not simple atomic actions, but rather long sequences of actions. Harvest is an example of a public goods game, where agents are tasked with collecting apples; however, apples respawn more frequently the more apples there are in nearby cells. Thus, if agents over-harvest the apples, they will no longer respawn. Cleanup is an example of a public goods game; here apples spawn more frequently when less waste has collected in the water channel at the top of the play area. Agents must clear the waste in order for apples to grow quickly, thus introducing the dilemma: who should clean the waste?

In order to apply RUSP to a grid world we add extra channels to the image the agent receives as observation. In cells containing another agent, we include the RUSP observations in these channels. We use a simple architecture: 1 convolutional layer (6 filters and a $3 \times 3$ kernel), flatten, 1 dense layer, 1 LSTM layer, layernorm, and finally any action heads. We used opensource code published with Jaques et al.[19]

In Figure 11 we show results when training with RUSP in these environments. We find that we are able to achieve comparable results to prior works,[17,23,19] though do not compare directly due to differences in optimization algorithms used. We found that the RUSP uncertainty was unimportant in these environments.

Figure 11: **Public Goods and Tragedy of the Commons.** Here we show renderings of the public goods games Harvest and Cleanup (taken from Jaques et al. [19]) and the collective reward for RUSP agents as compared to the greedy baseline. Consistent with past work, the greedy baseline fails to reach a solution with high collective return. RUSP agents, however, are able to reach cooperative equilibria with final collective return competitive with previously published methods. We found that adding uncertainty was unimportant in these environments, meaning that simply training agents with randomized prosocial but certain preferences was enough.

## C  Policy Learning

**Reinforcement Learning.** We consider the standard setting of stochastic multi-player games [54] where $n$ agents interact in an environment. Agents $1, \ldots, n$ are given a (potentially partial) observation of the true underlying state via an observation function $\mathcal{O} : \mathcal{S} \times \{1, \ldots, n\} \to \mathbb{R}^d$. Agents may each choose an action from their available set $\mathcal{A}^1, \ldots, \mathcal{A}^n$, and the next state $\mathcal{S}'$ is determined by a possibly stochastic transition function $\mathcal{T} : \mathcal{S} \times \mathcal{A}^1 \times \ldots \times \mathcal{A}^n \to \mathcal{S}'$. Finally, agents are rewarded as a function of the transition $\mathcal{R} : \mathcal{S} \times \mathcal{S}' \times \mathcal{A}^1 \times \ldots \times \mathcal{A}^n \to \mathbb{R}^n$. In MARL, agents are typically tasked with optimizing their own expected discounted future returns $R_i = \sum_{t=0}^{H} \gamma^t r_t^i$, where $r_t^i$ is reward for agent $i$ at time $t$ and $\gamma$ the discount factor. There are many algorithms to optimize this quantity; in this work we use Proximal Policy Optimization [43] and Generalized Advantage Estimation. [55]

**Optimization Infrastructure.** We use a distributed computing infrastructure used in Berner et al. [2] and Baker et al. [4]. CPU machines rollout the policy in the environment, compute the GAE targets and advantages, and send this data to the pool of optimizers. Optimizers (GPU machines) receive transitions and perform a PPO update and value network regression update. Periodically, new parameters are sent to CPU rollout machines, meaning that some amount of data is partially off-policy. In our work, samples are included in minibatches roughly 5 times, often called sample reuse.

**Past Policy Play.** During training we play against against past policy versions over the course of training. This has roots in double oracle algorithms [45] and has become standard practice in multi-agent reinforcement learning. [2,4,3] In this work unless noted otherwise, each agent has a 10% probability of being replaced by a policy sampled uniformly from all past versions. We found this to be extremely important in stabilizing emergence of both direct and indirect reciprocity.

**Policy Architecture.** We use a similar policy architecture to that used in Baker et al. [4] Each other agent is treated as an individual entity. The agent's own observations are concatenated with that of each other agent, they are individually embedded with 2 dense layers with 128 neurons, and then they are pooled across the entity dimension such that the result is a fixed size vector regardless of the number of agents. Finally, this fixed sized vector is passed through one more dense layer, an LSTM layer (both with 128 neurons), and finally layernorm [56] before action heads are computed. For action heads that attend over entities (e.g. in Prisoner's Buddy the choosing action head and in Oasis the action head that chooses which agent to attack), we concatenate the last activations after the layernorm layer onto each entity embedding prior to the pooling operations to form a new embedding

per entity. These are each then passed through an MLP with shared weights and output dimension 1, forming the logits per entity over which we take a softmax.

**Omniscient Value Functions.** Conditioning agents on varying social preferences is akin to training goal conditioned policies, and we therefore train a Universal Value Function[57] which is also conditioned on the reward transformation matrix. To reduce the policy gradient variance, we can make use of an omniscient value function giving it access to privileged information unavailable to the policy, common practice in both single-agent[58,59] and multi-agent[4,60,61] reinforcement learning. Agent $i$'s value function receives additional observations of the true (non-noisy) social preferences $T$, other agents' noisy observations over their social preferences $\widetilde{T}^{j \neq i}$, and other agents' uncertainty levels $\Sigma^{j \neq i}$.

**Optimization Hyperparamters.** Each training step is comprised of 60 gradient updates to both policy and value function parameters at which point new parameters are sent to rollout workers. Observations are normalized with an exponential moving average (EMA) mean and variance updated at each optimization step and clipped to be within 5 standard deviations of the mean. We also normalize the advantages per batch.

Unless otherwise noted below we use the following optimization hyperparameters.

| Batch Size | 8000 Chunks |
|---|---|
| BPTT Truncation Length | 5 Timesteps |
| Entropy Coefficient | 0.01 |
| Learning Rate | 3e-4 |
| Discount ($\gamma$) | 0.998 |
| GAE $\lambda$ | 0.95 |
| PPO Clipping | 0.2 |
| Normalization EMA $\beta$ | 0.99999 |

# D   Environment and Experiment Specific Details

## D.1   Iterated Prisoner's Dilemma

**Direct Reciprocity.** Agents observe both their own and their opponent's previous actions.

**Indirect Reciprocity.** Agents are endowed with an identity vector such that they can be distinguished from others in the game. On each episode, each agent is given a 16-dimensional identity vector sampled from $U[0, 1]$ which both they and their opponents can observe. For this experiment, we increased the horizon to 20 (as opposed to 10 for the direct reciprocity experiments) such that there is ample time for each agent to act as there are now 3. We also increase the back-propagation through time truncation length to 20 such that gradient signal can accurately propagate any information on reputation from prior timesteps.

## D.2   Prisoner's Buddy

Agents choose other agents with an entity-attention mechanism described in the previous section. Agents have a separate action head determining whether to sit out or to choose another agent, which is parameterized by a 2-class softmax. In the event that agent chooses to sit out, we mask gradients coming from the agent choosing action head (described in the previous paragraph) as these gradients will purely be noise.

Agents observe other agents' previous choices, and between rounds, agents are given 4 timesteps during which they can "choose" others but receive no reward such that they have time to break symmetry without penalty. Agents also observe who they and other agents chose in the previous round in which reward was given. We increase the mean horizon to 50, such that there are 10 rounds where reward is given.

## D.3   Oasis

The code for this environment was based off of that published with Baker et al.[4] Agents are spawned in random positions around the play area. They start with 20 health, which is the maximum health

they can reach. On each step, they lose 1 health. One food pellet spawns at a random position, and it can generate between 2.1 and 2.7 health worth of food each step (randomized per episode). If multiple agents try to eat the food on the same timestep, they will equally divide that health between them.

Agents can move by applying a force to their $x$ and $y$ axes and around their $z$ axis. At each step they may also choose one of three actions: eat, attack, or do nothing. If they choose to eat and there is food nearby, they will regain health. If they attack, they must also choose which other agent to attack via an attention mechanism. If that agent is close enough and in front of them, that agent will lose 5 health.

They observe other agents via an entity-invariant architecture that process state based representations of each other agent just as in Baker et al.[4] When an agent's health is reduced below 0, it "dies", receives -100 reward, and is sent to the edge of the play area where it held in place for 100 timesteps receiving no reward; after this period they are allowed to re-enter play.

The episode has 0.001% chance of ending on each timestep (a mean horizon of 1000 steps), and we also randomize the floor size on each episode between 1.5 and 6 which we found helped the early stages of training.

In this experiment we used a batch size of 32,000, which required 8 NVIDIA V100 GPU's and 2,000 CPUs over the course of approximately half a day. We also set the LSTM BPTT length to 100 timesteps and do not use any past policy sampling.