[Reviews · NeurIPS 2020]

Review 1

Summary and Contributions: The authors present a general approach to training agents that can resolve social dilemmas. Their method involves training the agents with randomized and uncertain social preferences. They demonstrate that even when evaluated with noiseless selfish preferences, the agents often exhibit stable solutions to social dilemmas. Specifically, they evaluate their method in the Iterated Prisoner's Dilemma, demonstrating both direct and indirect emergent reciprocity. They also demonstrate evidence for emergent team formation in a 5-player social dilemma variant of the PD they call "Prisoner's Buddy." They also provide preliminary evidence for team formation in a more complex MuJoCo-based game they call "Oasis."

Strengths: The authors provide a simple-to-implement and intuitive-to-understand approach to an important class of problems in multi-agent research. The paper is exceptionally well-written and presented. I'm no expert in IPD and social dilemmas, so it is difficult for me to judge how surprising / important these results will be to domain experts, as well as whether other baselines should have been included. However, I thought the results that were presented were interesting, surprising, and presented very clearly. What I found most exciting about this paper, however, is how generic and easy to implement the solution is. This should lend itself to other work building on top of it. I expect RUSP will be an important part of the story on multi-agent social dilemmas, but also potentially elsewhere as well. (See Relation to prior work.)

Weaknesses: I have only minor comments here. I think the paper is excellent and ready for publication as is. 1) The authors generally do a great job at providing intuitions behind their results, for example why uncertainty and information asymmetry are important. However, one thing I don't believe they commented directly on - why does cooperation persist in evaluations with selfish, certain preferences? One might have guessed that agents would learn during training that other selfish agents will defect, and therefore they should. Why do they not? Is it because they do not see agents trained with consistently selfish preferences? Is it because, since agents cannot observe the noise levels of other agents, there is still the possibility that the *other agents* are not certain of their selfishness and therefore may believe they are on a team and cooperate? 2) Section 5.2: The Oasis results were clearly preliminary and hopefully the authors will be able to update them for the camera-ready version. One thing in particular that bothered me: since all training setups tested results in similar mean reward, in what sense can we say that this social dilemma was "solved"? Sure, there is some evidence for team formation, but in what sense is that a "good" solution here?

Correctness: Only minor comments: 1) Section 3: Am I correct in understanding that the noisy T samples are not row-normalized like the true T? Should we be worried about this? Does it implicitly bias the agents in any way? 2) Figure 5: Perhaps I misunderstood the evaluation conditions, but with a past sample rate of 50%, doesn't that mean that 50% of the time, the last episode played is with a2, and therefore 50% of the time you are testing *direct reciprocity* rather than indirect? Is this really evidence for learned indirect reciprocity? This probably comes down to a fuzzy boundary between direct and indirect reciprocity in these conditions. To clarify, with the past play rate turned up to 100%, this would clearly test direct reciprocity only, correct? 3) Lines 224-225: Clearly, training with randomized social preferences helps here, but the effect of increased uncertainty looks less clearly significant to me. Could the authors provide a statistical test of the effect, or another visualization where the effect is more clear? Typos: 1) Figure 5 caption: 1st line has double "the" 2) Figure 11 caption: "We collection reward" - missing a "use" maybe?

Clarity: Generally, this was a very well written paper. I just have a few suggestions for clarifications: 1) Line 121-122: Agents observe relationships between others as well as to themselves? 2) Line 131: Including singleton teams? 3) Figure 3: Although it can be inferred from the plots and the main text, the mean horizon length should be explicitly marked on the figure or stated in the caption / main text, since it is important for interpreting the magnitude of the effect. 4) Figure 4: Isn't a beta distribution with beta=1 and alpha=100 basically just setting all block diagonal terms of Tij to 1? Why not just do/say that instead of mentioning the beta distribution? Is the tiny amount of variation around 1 useful? i would imagine its completely swamped by the noisy observations anyway. 5) Figure 7: For the lower right plot, a more forgiving metric might be the entropy of the death distribution. 6) Figure 11: Do the authors have any intuition as to what distinguishes cases where uncertainty over social preferences is important or not? 7) Figure 11: Typically for these environments, in addition to collective reward, people also look at measures of inequality. Could the authors provide some additional analysis to demonstrate what kind of solutions their agents are finding? For example, for Cleanup, are a couple of agents specializing in cleaning while others eat apples (high inequality) or are all agents sharing duties more equally (low inequality)?

Relation to Prior Work: The authors do a great job of discussing related work and placing their work in context, both narrowly and broadly. The only piece of related work I want to mention (not necessary to cite, but may be of interest to the authors) is Stuart Russell's "Human Compatible" book (and the related papers), which proposes that uncertainty over human preferences is an essential component of addressing the alignment problem.

Reproducibility: Yes

Additional Feedback: UPDATE AFTER REBUTTAL Thank you to the authors for their response. After further consideration on the calibration of my score against other accepted NeurIPS papers I've read, I have raised my score to a 9. Although I hope to see the paper accepted to NeurIPS, I look forward to reading the published version wherever it ends up!


Review 2

Summary and Contributions: This paper contributes a novel reward sharing strategy for multi-agent reinforcement learning. They show that it creates the right conditions for the emergence of reciprocity and indirect reciprocity in iterated prisoners' dilemma games. These results are important as they had long been hypothesized as possible but not directly shown before. The paper continues and shows how this method also works well in several new problems, including a complex new one which they introduce here called OASIS, and several widely used recent benchmarks (Harvest and Cleanup).

Strengths: The claims made in this paper are sound. The algorithm they propose is interesting, and broadly applicable within this field. I had a "why didn't I think of that!?" reaction after reading it. It's intuitively plausible why this approach would work. I also found the empirical evaluations to be convincing. The results on iterated PD are of deep theoretical significance. It has long been thought possible to get reciprocity to emerge from multi-agent RL on iterated PD, but had not been shown convincingly before.

Weaknesses: The OASIS part of the paper is both the most ambitious and the weakest (at least as currently presented). We are not given any real argument for why this environment has social dilemma-like dynamics. That's probably OK for a first paper on it though, especially since it's not the central point of this paper. The result is compelling enough as it is. Future work on OASIS should spend some time validating the environment, especially to show that it really is a social dilemma (if that's the intent for it). Another thing I would like to see more of in the paper is discussion of the limitations of the method. Are there any kinds of social dilemmas where the authors think it would fail? I also found the discussion of the evaluation setup for indirect reciprocity, with the hold-out and rapport protocols, to be rather confusing. A small amount of effort in improving the readability of this section might yield a big effect on the eventual impact of the paper. Like the result on direct reciprocity, this result may also be highly significant. It should be explained more clearly.

Correctness: yes

Clarity: yes

Relation to Prior Work: yes

Reproducibility: Yes

Additional Feedback: Minor: - What does the X-axis of the Harvest and Clean-up figures in the appendix mean? It is just labeled "training iterations". - The notion of a social autocurriculum appears uncited in the broader impact statement, it might be helpful to add a reference.


Review 3

Summary and Contributions: This work presents an interesting training method to resolve social dilemmas. The authors propose to train agents with randomized uncertain social preferences (RUSP) by randomizing reward function. To learn socially reactive behaviour, they induce asymmetric uncertainty over agents’ social relationships. It has been shown that training with RUSP results in emergent direct reciprocity, indirect reciprocity and reputation, and team formation. A number of experimental results on different games show that learning these behaviours with RUSP results into higher social welfare equilibria. The proposed method outperforms a selfish baseline method defined by the authors. POST REBUTTAL: I have read the authors response and they have addressed most of my comments. I believe the proposed method is a novel way to resolve social dilemma and agree with other reviewers that the approach is intuitive. Authors justification of not using other baseline methods has kind of convinced me but I still feel that they could use baselines such as inequity aversion/reward shaping for few of the experiments. Reviewer 4's comment on theoretical grounding is also a concern. Overall, I am increasing the score to 5.

Strengths: The paper proposes to randomize reward matrix and include uncertainty in the agents' relationship which I think is a novel way to resolve social dilemmas. Resolving social dilemma and learning high social welfare equilibria is of importance to many real world applications and people working on game theory will find it relevant. The main highlight of the paper is its experimental section. Experiments have been performed on a number of games to show strength of RUSP in learning reciprocity and team formation behaviours. The results show that these bahviours results in learning cooperative equilibria which provides higher social welfare values.

Weaknesses: - The clarity of paper can be improved (see below for detailed comment) - I feel the work is closer to game theory literature than MARL literature. - An algorithm or clear steps of the method would help understand the proposed method clearly. - Showing computation of reward transformation matrix for one of the games used for experiments will be helpful. - My major complaint is the proposed method has not been compared with any other work. Authors have talked about a number of existing work to resolve social dilemma (ex. inequity aversion, reward shaping etc.), however they compare their work only with a selfish baseline of the proposed method. Comparing the work with other existing work will improve the paper.

Correctness: The proposed method and empirical methodology appears correct to me.

Clarity: I believe the writing and flow of the paper can be greatly improved. The content and section names do not match. In first read it was not clear to me that section 4 and onwards are experiment sections. Follwing are some points which I feel should be addressed to improve the readibilty of the paper - Preliminaries section looks more like literature review section. Social dilemma and team formation problem should be explained more clearly. - The actual method description started only in the third paragraph of method section. It would be helpful if the method was introduced first and then discuss relevant steps/parts. Also, having an algorithm would help readers understand it better. - It was not clear what should I expect from a few of the sections. Setting the expectation at the start of each section improves reading experience. Similarly, readers should know that they are reading experiment section.

Relation to Prior Work: The authors have discussed work related to resolving social dilemmas and have mentioned that the existing work are either not scalable to large number of agents or do not exhibit emergence of reciprocity. They have also discussed work related to resolving team formation dilemmas. Overall, literature review seems adequate to me.

Reproducibility: Yes

Additional Feedback:


Review 4

Summary and Contributions: The authors propose a novel environment for training multi-agent simulations with randomized uncertain social preferences (RUSP). Key features are (i) random reward transformation matrix that transform pre-existing reward vectors; (ii) information asymmetry as agents have independent uncertainty regarding their relationship with each other; and (iii) the RUSP distribution contains the unmodified selfish environment, with no social preferences or uncertainty, so agents can be evaluated without any social preferences. The authors show that their RUSP agents establish cooperative equilibria and demonstrate emergent reciprocity, shown on iterated Prisoner’s dilemma, and team formation, shown on Prisoner’s Buddy and the MUJOCO Oasis environment. The authors show in their experiments that even agent teams with high uncertainty develop reciprocity. In fact, reciprocity emerges more quickly under high uncertainty, and never emerges with no uncertainty.

Strengths: The proposed random transformation matrix offers a simple mechanism with which to encourage cooperative agents. The authors offer extensive evaluations, demonstrated on both matrix games and also complex physics-based environments, and clearly show the emergence of reciprocity and teamwork. The authors will make their multi-agent environments open-source, thus more useful to the broader community.

Weaknesses: Emergence of cooperation is a well-studied area. In proposing this RUSP environment augmentation, the justification for this approach is not fully convincing. Is there grounding in literature either in psychology or animal behavior (e.g., termites, bees) that suggest information asymmetry and random rewards? Are there previous human subject studies in the lab for example that could support RUSP environment augmentation? The authors do not offer a theoretical grounding for their work. The empirical evaluation is also limited. For example, in evaluating Infinite Prisoner's Dilemma, they only consider one configuration of the reward matrix. In the experiments, the authors do not describe their RL methods; it is unclear which RL algorithms they leveraged. Additionally, the authors could include naive baselines other than simply a selfish baseline (which by design would result in a large number of agents defecting).

Correctness: The claims are correct. The empirical methodology is correct, although could be more comprehensive.

Clarity: The paper overall is well written. Line 76: missing word "we consider the problem of team formation a social dilemma" Notation such as R, P, S, T are introduced in the caption of Figure 1; they should be instead (or also) introduced in the body of the text for clarity. Please cite the peer-reviewed version of papers when they exist instead of arXiv versions. For example, [1] should be the Science 2018 paper, not arXiv 2017. References also seem to be in a non-standard format (as superscripts, as opposed to e.g., [1]) making them harder to see.

Relation to Prior Work: Yes, this paper shows for the first time a multi-agent RL that exhibits emergent reciprocity and team formation. They ground their paper in past work on social dilemmas, cooperative equilibria, and team formation.

Reproducibility: Yes

Additional Feedback:

[Author Response · NeurIPS 2020]

We thank the reviewers for their time and thoughtful feedback! We are encouraged that they found our method intuitive (**R1**,**R2**), simple (**R1**,**R4**), novel (**R2**,**R3**,**R4**), well positioned with respect to past work (**R1**,**R2**,**R4**), and overall clearly presented (**R1**,**R2**,**R4**). We were furthermore heartened that reviewers found our empirical evaluations were a highlight (**R3**), extensive (**R4**), convincing (**R2**), and surprising (**R1**). We found the criticisms overall very constructive and appreciate them greatly regardless of whether our submission is accepted!

**More baselines** The most common criticism from the reviewers was that we only experimentally compare our method to purely selfish agents (**R1**,**R3**,**R4**). We present the first method that shows emergence of *both* reciprocity and team formation, and because there are no other methods that have claimed as such, we felt there were no obvious choices with which to directly compare. High social welfare, robust equilibria have been so elusive that MARL social dilemma research often investigates whether these behaviors can emerge *at all* rather than comparing efficiency in obtaining those behaviors. Finally, none of the reviewers suggested specific methods with which to compare, and comparing against the multitude of prior methods that achieve only a subset of the behaviors emergent with RUSP would be very labor and compute intensive to the extent it may deserve its own independent publication and so we chose to leave this to future work.

**R1**- **More intuition on sustained cooperation** Thank you for these great questions! We are happy to add more discussion around this in the paper. Our intuition on why cooperation persists to evaluations with selfish, certain preferences is that there are cases during training where agents with selfish preferences but asymmetric uncertainty (such that one has selfish, certain preferences but the other selfish, uncertain preferences), allowing the agent to experience the requisite variance over cooperative and defective strategies. In cases where agents learn without uncertainty during training, we believe it may be from the smooth transition over the threshold where cooperation is directly incentivized (high reward sharing) and where it is less clear if it is beneficial (low reward sharing). This hypothesis is somewhat supported by the results in Figure 4 where we see the cases with hard teams and no uncertainty failing to learn cooperative behavior.

**R2**- **More discussion of method limitations** We are happy to discuss limitations more in the paper. We already mention the potential credit assignment issue with reward sharing methods in Section 6; we also found that past policy play was necessary and would like to investigate more in future work how this and other methods that induce variance in agent play interact with RUSP.

**R3**- **"I feel the work is closer to game theory literature than MARL literature."** Works like ours that focus on learning in higher complexity environments such as harvest, cleanup, and oasis have historically been submitted to ML journals rather than game theory journals. The focus of this paper is providing pressures for MARL methods to converge to higher social welfare equilibria so we feel it belongs in the MARL camp.

**R3**- **"Showing computation of reward transformation matrix for one of the games used for experiments will be helpful."** We already give an example of this in Figure 2(b).

**R4**- **"In proposing this RUSP environment augmentation, the justification for this approach is not fully convincing."**, **"The authors do not offer a theoretical grounding for their work."** Our motivation (in Section 3 of the paper) is based around the intuition that RL agents will learn adaptive strategies in the face of uncertainty or partial observability. In RUSP, we induce uncertainty over social preferences which we hypothesize and validate experimentally leads to social adaptability and robust cooperation. That being said, we would love to see future work linking RUSP to biological mechanisms or more game theoretic justifications for the method, and we will mention this as an interesting avenue for future research in our discussion section.

**R4**- **"The empirical evaluation is also limited"** Evaluating in multiple IPD matrices is not standard practice to our knowledge (e.g. see LOLA); in general modifying the payouts should simply move the threshold on the discount factor at which some cooperative strategies such as Grim Trigger become Nash. We did not cherry-pick or finetune this matrix – it was simply the first one we tried, and we are happy to say as such in the paper.

**R4**- **"the authors do not describe their RL methods; it is unclear which RL algorithms they leveraged"** We describe them in the Appendix (C) and say as such on L147. Our RL algorithm (distributed PPO with omniscient value function) is standard so we did not think it important enough to describe in detail in the main text but will add more.

**Improving Paper Clarity** In the final submission we will be happy to include algorithm pseudo-code (**R3**), add introductory sentences to the sections to set expectations (**R3**), expound upon the problem definitions in the Preliminaries section (**R3**), divide the Method section into subsections "motivation" and "method" (**R3**), add more discussion about the Oasis environment and results (**R1**,**R2**), clarify the explanation of the indirect reciprocity evaluation setups (**R1**,**R2**), soften language around uncertainty's efficacy for Prisoner's Buddy (**R1**), and fix any typos + minor clarifications (**R1**,**R2**,**R3**,**R4**). We thank the reviewers again for the their time and thoughtful comments, and we hope that with these (and those listed above) improvements to the paper, **R3** and **R4** will consider increasing their score!

[Meta-Review · NeurIPS 2020]

Four knowledgeable referees reviewed this paper. After conducting initial reviews, reading the authors’ rebuttal (which resolved some concerns, but not the core concerns of two of the reviewers), and discussing the paper, the reviewers did not agree on an outcome. Two of the reviewers came to the conclusion that this is a ground-breaking paper (simple and elegant). The other two reviewers were perhaps somewhat intrigued, but did not feel the paper was yet ready for publication. For example, during the discussion phase, R4 (a very accomplished and well-respected research in the field) made very valid points about the papers weaknesses: “So all this leads me to suggest that there needs to be a better context, more related work and a better way to situate the paper in related arenas, e.g., provide some sort of a framework to back up the findings. I understand the issue of limited space, but given the amount of literature in this area, I feel that the paper doesnt do a good enough job explaining its findings in context.” and “This paper is different from other topics (e.g., papers looking at fairness in ML) in that this topic of pro-social behaviors in PD/IPD has seen decades of previous work including in multiagent systems, it is important in my view to situate this work better in context and provide more theoretical basis. The Kahn & Murnighan (1993) paper also discusses uncertainty in rewards. There are other papers that have discussed such uncertainty in PD/IPD such as the one I mentioned.” I read the paper in detail, and considered the reviewers’ comments and the authors’ rebuttal. I can see how experts can have differing opinions about the extent and validity of the paper’s contributions when viewed from different perspectives. From the perspective of getting deep RL algorithms to cooperate (including various forms of reciprocity and “team formation”) in repeated prisoner’s dilemmas, this potentially represents a nice achievement. And RUSP *seems* to be simple and quite compelling (though it isn’t at all clear to me how robust it is). The two reviewers in favor of the paper rightly, from this perspective, appreciated the success of the method in producing compelling behavior in several domains. Yet, not everything is about deep RL, which repeatedly been shown to be a tool with extreme limitations in multi-agent reinforcement learning outside the context of zero-sum games. As R4 points out, a study of how rational agents learn cooperative behavior (and reciprocate, form teams, etc.; which has been studied extensively in many disciplines ,including AI and the NeurIPS community, for many years) is not just about current RL methods. Thus, when the paper is viewed from this broader perspective, it is (though interesting) somewhat dissatisfying. The approach does not seem to be theoretically justified, nor do the results presented confirm the claims (they provide some evidence, but they do not thoroughly evaluate the strengths and weaknesses of the approach nor its robustness). Thus, from this perspective, reviewers can easily worry that by accepting the paper, it could open up a firestorm of misinformation that could potentially side-track the progress of the field. Overall, I think the approach and results described in the paper are compelling and could have a good impact on the field. So I believe that it could be accepted at NeurIPS. That said, I hope the authors will exhibit honesty and care as they present and ground their claims and results in the final version of the paper. In particular, I strongly urge the authors to provide satisfactory *context* and *a theoretical basis or argument* for their approach and results. Without such context and theoretical basis, the paper risks coming across as unprincipled hackery. [In saying all this perhaps overly bluntly, I do not wish to demean the paper's approach or results in any way. I simply hope the authors will take all of the reviewers' comments seriously in order to improve the final version of their paper.] ==== A couple of other points I had after reading the paper: - Like R1, the Oasis results do not look good to me, and seem to indicate the method might not achieve what is hoped. The mean reward and total deaths seem the same in all games. I didn’t quite understanding the authors’ conclusions from the results presented. R1 noted that they hoped the authors would improve these results (though that is a somewhat worrisome comment to me, as it suggests a need to fine-turn and tweak results until the desired outcome is achieved rather than demonstrating generalizable knowledge). - Some of the presentation is confusing. I’m not sure what a “training” iteration represents, nor was the setup of the games fully clear to me. Perhaps I just missed the explanations in the main paper or it is explained in the appendix (a description in the main paper seems desirable to me, though I understand space limitations).